# Incidence Trends and Main Features of Gastro-Intestinal Stromal Tumours in a Mediterranean Region: A Population-Based Study

**DOI:** 10.3390/cancers15112994

**Published:** 2023-05-30

**Authors:** Ricardo J. Vaamonde-Martín, Mónica Ballesta-Ruiz, Antonia Sánchez-Gil, Juan Ángel Fernández, Enrique Martínez-Barba, Jerónimo Martínez-García, Gemma Gatta, María D. Chirlaque-López

**Affiliations:** 1Service of Epidemiology, Region of Murcia Health Council, Ronda de Levante 11, 30008 Murcia, Spainmdolores.chirlaque@carm.es (M.D.C.-L.); 2Institute for Biomedical Research of Murcia, IMIB-Arrixaca, 30120 Murcia, Spain; 3School of Medicine, University of Murcia, 30100 Murcia, Spain; 4SMS (Region of Murcia Health Service) Calle Central, 7, 30100 Murcia, Spain; 5University Hospital “Virgen de la Arrixaca”, 30120 Murcia, Spain; 6Evaluative Epidemiology Unit, Fondazione IRCCS Istituto Nazionale dei Tumori, 20131 Milan, Italy; 7CIBER Epidemiología y Salud Pública (CIBERESP), 28029 Madrid, Spain

**Keywords:** GIST, malignant, population-based, survival, cancer, incidence

## Abstract

**Simple Summary:**

A Gastro-Intestinal Stromal Tumour (GIST) is a rare type of cancer discovered in the last decades of the 20th century and whose characteristics are still being defined. Knowledge of the distribution of any disease and its severity among a population is paramount for the best distribution of efforts and resources intended for its diagnosis and treatment. The present study was started by the European Union Joint Action on Rare Cancers and is aimed at shedding light on the current status of GISTs in a region in the southeast of Spain, the matter being quite unknown in the whole of Europe because of there being few previous studies and partial invalidation of those studies due to prior consideration of some subgroup of GIST as non-cancerous. Aside from producing useful information for health authorities, our study will contribute to expanding the knowledge of this relatively new and, from a population level point of view, poorly known cancer.

**Abstract:**

Gastro-Intestinal Stromal Tumours (GISTs) are a kind of neoplasm whose diagnosis in common clinical practice just started in the current century, implying difficulties for proper registration. Staff from the Cancer Registry of Murcia, in southeastern Spain, were commissioned by the EU Joint Action on Rare Cancers into a pilot study addressing GIST registration that also yielded a population-based depiction of GISTs in the region, including survival figures. We examined reports from 2001 to 2015 from hospitals as well as cases already present in the registry. The variables collected were sex, date of diagnosis, age, vital status, primary location, presence of metastases, and risk level according to Joensuu’s Classification. In total, 171 cases were found, 54.4% occurred in males, and the mean age value was 65.0 years. The most affected organ was the stomach, with 52.6% of cases. Risk level was determined as “High” for 45.0%, with an increment of lower levels in recent years. Incidence for the year 2015 doubled that of 2001. Overall, the 5-year net survival estimation was 77.0%. The rising incidence magnitude is consistent with trends in other European countries. Survival evolution lacked statistical significance. A more interventional approach in clinical management could explain the increase in the proportion of “Low Risk GISTs” and the first occurrence of “Very Low Risk” in recent years.

## 1. Introduction

GISTs (Gastro-Intestinal Stromal Tumours) represent a clinical entity that began to be defined by the end of 20th century [1] and whose diagnosis in common clinical settings started in the very first years of the 21st century [2]; until then, these tumours were misdiagnosed as other types, mainly leiomyomas and leiomyosarcomas. Through the first two decades of the present century, the diagnosis of GIST cases has been increasing in a strong, steady manner due to the implementation of several immunohistochemistry tools [3]. The introduction of targeted therapies, which brought a much better prognosis for disseminated or recidivated GIST disease [4,5,6], also contributed to a more refined differential diagnosis process.

As a consequence, during the last two decades there has been an important increment in GIST knowledge and an ongoing debate about the need to characterize all GISTs as malignant entities; this point of view has faced opposition by the fact that a significant portion of GISTs were misdiagnosed as leiomyomas in the past, which are well known for their true benign nature. Regarding this, the WHO, in the draft of the second revision of the *International Classification of Diseases for Oncology*, third edition (ICD-O-3) [7], has proposed classifying all GISTs as malignant neoplasms.

On the other hand, GISTs stand out in the classification of rare tumours stated by the European Union Joint Action on Rare Cancers (JARC), RARECARE project [8], as one of the four big classes that make up the sarcoma family. Its consideration as a “rare cancer” is due to its low annual incidence, less than 6 cases per 100,000 inhabitants, for Europe as a whole [9].

The uncertainties due to very low incidence and rapidly evolving conceptions about its nature and prognosis prompted the creation of a working group framed into the JARC whose purpose was to improve the registration and quality indicators of GISTs. To share the knowledge gathered by the Cancer Registry of the Region of Murcia (RCM) during its participation in the above-mentioned working group, an epidemiological analysis at a population-based level of GISTs was promoted. Thus, incidence, temporal trend, estimated risk of malignant behaviour according to pathology reports, and survival of GISTs in a Spanish population were the aim of the present study.

## 2. Materials and Methods

The study design was a retrospective cohort using data from the RCM, a population-based cancer registry that gathers incident cancers occurring among people that have their residence in the Region of Murcia in southeastern Spain. During the period of study, the region’s population increased from 1,190,378 people in 2001 to 1,467,288 in 2015 [10]. The coverage of the registry is up to 95%, and the information comes from discharge summaries from hospitals, pathology reports from public hospitals and private laboratories, and also information from the official mortality statistics. 

The RCM has been publishing data periodically in the Cancer Incidence in Five Continents (CI5) monographies [11] and is a member of the European Network of Cancer Registries (ENCR) [12] and the Spanish Network of Cancer Registries (REDECAN) [13,14].

A review was conducted on all the cases already registered with the morphology code “8936” (GIST) (according to ICD-O-3) in the RCM during the incident period 2001–2015. Additionally, because recommendations were made prior to the JARC [15], many GISTs were considered non-registrable tumours, so the reports from pathology, oncology, and radiotherapy services and hospital discharge summaries from the same period were examined.

The variables collected were sex, date of diagnosis, age at diagnosis, presence of other cancer(s), existence of metastatic extension of GISTs within the first 6 months after diagnosis, primary site, and data from pathology reports in order to determine the recurrence/progression risk level according to Joensuu’s Classification (tumour size, primary site, number of mitosis per 5 mm^2^, and presence of tumour rupture) [16]; although originally proposed by Joensuu to ascertain the risk of recurrence/progression after total excision of a localized GIST, in the present study it was explored just for risk of death as a consequence of the GIST. Metastatic disease at diagnosis was considered as worse than tumour rupture from the perspective of assessing risk of death; therefore, a risk level higher than Joensuu’s high risk level, named “very high risk”, was assigned to metastatic cases without the need to take into account all pathology information from the primary tumour.

Vital status at the end of follow-up period and date of death were collected from official death registry and clinical registries. Each case was followed for a minimum of 5 years. The follow-up period did not include collection of other events apart from death.

### Analysis

Crude and adjusted incidence rates standardized to 2000 Standard World Population (SWP) and 1976 and 2013 European Standard Population (ESP) were calculated; 95% confidence intervals (95% CI) for the adjusted rates were estimated by the Tiwari method [17], as recommended by the ENCR for rare cancers. The year of diagnosis ranged from 2001 to 2015 and was subdivided into three periods (quinquennia): 2001–2005, 2006–2010, and 2011–2015.

For evaluation of possible differences by sexes, Pearson’s chi-square and Fisher’s tests for tumour site and death risk level and Student’s t tests for age were performed.

The analyses were conducted separately for sex, age, period, and risk group. Differences in incidence rates were evaluated using Standardized Incidence Rate Ratios (SIRs).

Observed survival (OS) was estimated by the Kaplan–Meier method, stratified by sex, period, age, risk level, and primary site of tumour; analyses were performed only for strata that have more than 10 cases. Calculation of net survival at 1, 3, and 5 years (including their 95% CI) was performed via Pohar-Perme method, where net survival is defined as the survival under a hypothetical situation where the only cause of death would be GIST [18,19].

Differences in survival between sexes, periods, age groups, and death risk level were evaluated using a multivariate flexible parametric model survival approach to assess the excess mortality hazard ratios (EHRs) and 95% CIs through restricted cubic spline-based hazard models with three knots [20].

The end of follow-up period was dependent on year of diagnosis, but a minimum of 5 years of follow-up was granted. Lifetables used have been the official ones for the Region of Murcia since year 2000 to 2020, where reference population was sourced from the official yearly registry of residents in the Region of Murcia, detailed by sex and age [10].

For specific analysis, “low” and “very low” Joensuu’s risk levels were grouped together because of the low number of cases in the latter and proximity in their biological behaviour (in fact, post-surgery management is equal in clinical practice for these groups) [21]. Statistics were evaluated using 0.05 significance level two-tailed tests.

The statistical software Stata v.14 was used (StataCorp LP, College Station, TX, USA). 

The study was conducted in accordance with the EU 2016/679 General Data Protection Regulation (GDPR) regarding the use of anonymised population data [22]. All data collected in the study database for analysis were anonymous; thus, no further ethical approval was required.

## 3. Results

In total, 171 cases that had confirmation of diagnosis by microscopy in the period 2001–2015 were included, consisting of 93 men and 78 women (Table 1); incidence was significantly higher in men (*p* = 0.032). All cases had enough information in their pathology report to be categorized in an accurate way into one of the five risk levels. In all the cases, it was possible to ascertain the vital status for a minimum of 5 years (no loss of follow-up occurred), the mean follow-up time being 5.5 ± 2.9 years.

The most reliable sources for diagnosis were histologic analysis of the primary tumour in the great majority of cases (n = 163); in five cases, only the histology of metastasis was available and just three cases had been diagnosed based on material from cytology only. In all the cases, microscopy was supported by the use of immunohistochemistry or molecular biology techniques. Mean age for the whole set of cases (Table 1) was 65.0 years with a median of 67.5, and it was higher in women (66.9 vs. 63.4 years), but differences lacked statistical significance (*p* = 0.100 for mean age and *p* = 0.194 for median). Regarding tumour site, the most frequent organ involved was the stomach, with 90 cases; differences between sexes in tumour site were not statistically significant (*p* = 0.088). Risk level for death was distributed as follows: 31 cases (18.1% of total) were classified as very high risk, 77 (45.0%) as high risk, 22 (12.9%) as intermediate risk, 26 (15.2%) as low risk, and 15 (8.8%) as very low risk, without statistically significant differences by sex (*p* = 0.707) (Table 1).

Thirty-six cases also had another malignant tumour, eleven of them had the other cancer diagnosed previous to the GIST, fifteen had synchronous tumours, eleven had the other cancer diagnosed after the GIST (one of the cases had both previous and synchronous). The distribution of these associated cancers was not homogeneous among the different risk groups: the percentage of low/very low risk tumours associated with other cancers yields a 3.9 to 1 ratio when compared to those in the intermediate/high/very high risk groups.

The annual crude incidence rate of GISTs per 100,000 inhabitants was 0.82 overall, with a value of 0.88 in males and 0.76 in females; the difference was statistically significant (*p* = 0.032). The standardized age-adjusted rates for both sexes ranged from 0.53 when adjusted to SWP to 1.05 for ESP-2013, with a value of 0.75 for ESP-1976 (Table 2).

When grouping incidence values in averaged data for each 5-year period from 2001 to 2015, the values were similar for the first two quinquennia (adjusted rates to ESP-76 of 0.54 and 0.52, respectively), while for the last quinquennium it turned out over twice as much (1.15). This temporal trend in incidence was similar when considering each sex separately. The same temporal evolution was observed for the proportion of lower risk (low + very low) GISTs: there was a relevant, statistically significant (*p* = 0.006) increase in the last quinquennium (2010–2015): 33.7 % vs. 9.1 and 7.5% in previous periods.

The standardized incidence ratios of GISTs (Figure 1) were lower for the intermediate Joensuu’s risk group, while higher in men and for the last period as compared to the first (2011–2015 vs. 2001–2005).

The overall observed survival at 5 years was 68.4%, with a value of 62.4% for men and 75.6% for women. The maximum survival figure by subgroups was 90.9% for intermediate Joensuu’s risk level, while the minimum value (48.4%) was observed in the group of people with very high risk level. Detailed information can be found in Figure 2 and Table 3.

Overall, 5-year net survival (Table 3) was estimated at 77.0%, being consistently better for women than men and also for younger people than older people, but the differences were not statistically significant.

Regarding net survival by risk groups, values were slightly better for the intermediate risk group, without reaching statistical significance in the multivariate analysis (*p* = 0.333) compared to the reference group (high risk). Only in the very high risk group, which had the worse net survival figure (52.7%), the hazard ratio difference was statistically significant (*p* = 0.047).

Table 4 shows the risk of dying from a multivariate flexible parametric model, and the risks are reciprocally adjusted for sex, age, period, and risk level. The highest risks of death are for the very high risk level, males, and people aged more than 64 years, although just one of the hazard ratios reached statistical significance: very high risk, *p* = 0.047 (*p* = 0.091 for sex; *p* = 0.292 and *p* = 0.096, respectively, for age group 40–64 years and more than 64 years compared to the younger group).

## 4. Discussion

This is one of the few European studies showing the analysis of all GISTs, including also those formerly considered “benign, and according to the risk stratification. In our study, GISTs’ incidence is on the rise in the Region of Murcia, to the extent that estimates for 2015 more than double those of 2001. During the 2011–2015 period, there has been an increase in the proportion of GISTs that qualified as “low risk” and, for the first time in our series, there appeared cases of “very low risk”. Survival has shown a positive evolution throughout the fifteen years evaluated, although the results do not reach statistical significance. 

The incidence trend in our study is concordant with temporal trends observed in other European regions or Western countries [23,24,25,26,27,28,29,30], where an increment through the years is always present, although not statistically significant in every instance. This increment should be regarded as somewhat artefactual because GISTs in general, but especially those of “low” and “very low risk”, would have been heavily underreported because there was a lack of awareness of the need for surgical removal of these lesions even in absence of clinical manifestations. Considering only the most recent annual incidence rates available from these trend studies (among the various existent for each year or period), adjusted to the 1976 European Standard Population or the 2000 United States of America Standard Population, we observed that they range from 0.44 in the Czech Republic and Slovakia to 1.30 in Germany, with values around 1.0 for France, Italy, the Netherlands, and Spain (1.15 in our study) and 0.70–0.78 for the USA; these values are higher than the estimation from Rarecarenet [8] for the whole of Europe (0.30), but that is a mean value extracted from the 2000 to 2007 period and when only malignant lesions were collected. During that time period, the incidence increase was the highest compared to the other rare cancers with an annual percentage change of 24% [31]. Furthermore, differences between European regions were important (from 0.05 to 0.5) [8]. It must be noted for comparability that we took into account low and very low risk GIST cases while many of the population-based studies have not accounted for the lower risk groups of GIST. Other conclusions that can be consistently obtained from these studies (and ours) are a fairly equal distribution by sexes and a peak incidence around the seventh decade of life. These results were also confirmed by the Rarecarenet project based on slightly less than 5000 cases, therefore reaching statistical significance. 

The shift in proportion sharing in favour of low-risk lesions is concordant with the improvement in preoperative diagnostic capacities [32] and rising awareness among surgeons of the importance of removing all incidental lesions suggestive of a GIST, whatever the size, when occurring outside the stomach; only in the latter organ, surgical risks outweigh the benefits when the lesion is less than 2 cm [33,34,35,36]. 

Regarding prognosis, the increase in incidence has fortunately come along with a huge increment in knowledge applicable to diagnosis, treatment, and follow up, as is well summarized in the ESMO guidelines from 2018 [21]. Our results contribute survival data of GISTs to previous European studies performed in the general population or in clinical cohorts [23,30,37,38,39]. Survival rates have shown a positive evolution through the first decades of the present century in those population-based studies that addressed this issue [25,40,41,42]; this is also true for our data, but they lack statistical significance. Some studies [23,43,44], in line with ours, find an inexistent or even reversed difference in survival between low, very low, and intermediate risk level cases. This paradox can be explained by some factors: first of all, the fact that Joensuu’s and other risk levels were established with data generated prior to the widespread use of imatinib, people then being treated with surgery alone; additionally, in our study, low/very low risk tumours were more frequently associated (19.5% of the cases) with other cancers, the GIST probably being an incidental finding during staging or treatment of the other cancers (that were likely to be a competitive risk of death). The percentage of low/very low risk tumours associated with other cancers is almost four times that of the intermediate/high risk group. Deaths during the first month after surgery were also more frequent in the low/very low risk groups, but to a lesser extent (7.3% vs. 4.3%), implying a greater degree of comorbidities as the most probable explanation.

When considering gender differences, we have found greater incidence in men than women, that difference being the only one significant in statistical terms and confirmed by the European data [8]. On the contrary, differences in 5-year survival and other small differences in tumour site, risk level, age at diagnosis, and stage at diagnosis did not reach statistical significance. Studies in other European locations considering this aspect show variable results, the difference always being a small one. According to the European data, survival was significantly better for women than men and for younger (<65) than older (65+). Furthermore, survival increased rapidly from 1999–2001 to 2002–2004 and then remained stable until 2005–2007.

Our study is the first to describe the incidence of GIST cases according to patient characteristics and survival in the Region of Murcia from a population-based perspective. In this aspect, it is also a pioneer in Spain, where there are very few regions that account for similar scientific works. A major advantage of being population-based is that it includes virtually all patients with GIST, minimizing the selection bias. The good information available in the pathology reports of our region must be noted, which allowed us to categorize all the available cases into their proper Joensuu’s risk level. For doing so, we also had to overcome the issue of accurately defining the “tumour rupture” concept, which has been troublesome for everybody since its proposal as a risk factor [45,46]. We have to praise pathology reports that not only provided us with accurate information for the study, but in many cases, they contained gene mutation analysis that was very useful in the selection of better targeted therapy for each patient [21,47,48].

Limitations of the study are mainly derived from the conjunction of an incidence in the range of rare cancers and a relatively limited size of the population in our region, yielding an insufficient number of cases to reach statistical significance in most of the analyses. Moreover, a lack of access to the whole clinical history precludes multivariate analysis that would have confirmed comorbidities and multiple cancers as the cause of similar survival figures in the low/very low risk group compared to high-risk cases.

## 5. Conclusions

For the first time, we displayed GIST patients’ clinical and epidemiological characteristics and survival trends in a southeast Spanish region in a population-based context. This study highlights the pertinence of registering the GISTs of low and very low risk of recurrence, because nowadays they are also subjected to treatment and therefore must be evaluated in order to know the consequences of that strategy. Though we found general survival improvements, they lacked statistical significance, and when comparing certain subgroups’ survival figures they seemed quite counterintuitive; this warrants further investigation for a better understanding of GIST prognosis according to clinical and epidemiological factors.

## Figures and Tables

**Figure 1 cancers-15-02994-f001:**
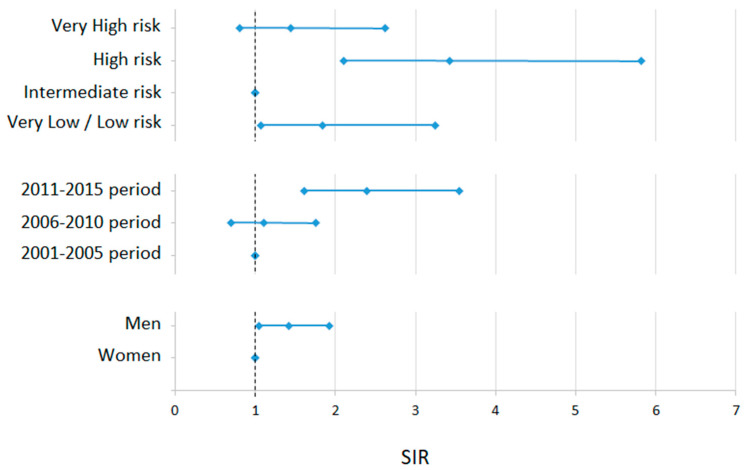
Standardized Incidence Rate Ratios (SIRs) (and their 95% CI) between subgroups of GIST patients. Subgroups evaluated are Joensuu’s risk level, period of diagnosis, and sex. Reference categories are those with value = 1.

**Figure 2 cancers-15-02994-f002:**
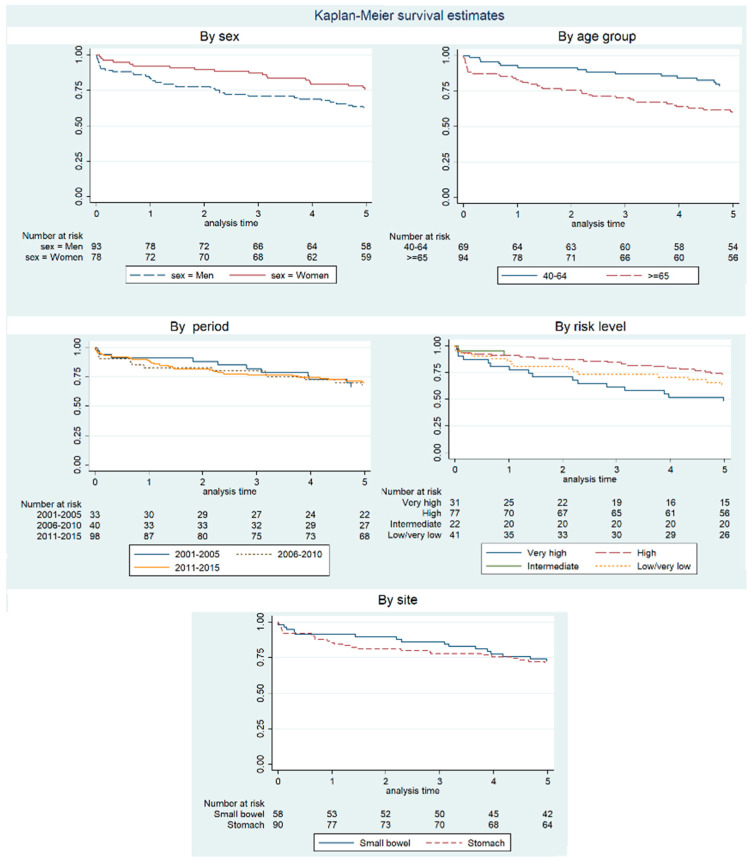
Kaplan–Meier survival curves of OS by sex, age period, level risk, and site group of GIST patients. Because of scarcity of cases, only the two site groups with more incidences and age groups of 40 years and over are presented.

**Table 1 cancers-15-02994-t001:** Gastro-Intestinal Stromal Tumour cases distribution by sex, age group, primary site, and Joensuu’s risk level. P value of tests performed show statistically relevant differences by sex. Period 2001–2015, Region of Murcia, Spain.

	Cases			
	Total	Men	Women	*p*-Value
	N (%)	N (%)	N (%)	
**Total 2001–2015**	171 (100%)	93 (54.4%)	78 (45.6%)	0.032
**Period**				
2001–2005	33 (19.3%)	19 (20.4%)	14 (18.0%)	0.792
2006–2010	40 (23.4%)	20 (21.5%)	20 (25.6%)	
2011–2015	98 (57.3%)	54 (58.1%)	44 (56.4%)	
**Age groups, years**				
15–39	8 (4.7%)	5 (5.4%)	3 (3.9%)	0.176
40–64	69 (40.3%)	43 (46.2%)	26 (33.3%)	
65 and older	94 (55.0%)	45 (48.4%)	49 (62.8%)	
**Age, years**				
Minimum age	24.5	24.5	32.0	
Maximum age	89.2	88.5	89.2	
Mean age	65.0	63.4	66.9	0.100
95% confidence interval for mean age	62.9–67.1	60.4–66.5	64.1–69.6	
Median age	67.5	64.6	69.0	0.194
**Primary site of tumour (CIE-O)**				
Oesophagus	3 (1.8%)	2 (2.1%)	1 (1.3%)	0.088
Stomach	90 (52.6%)	45 (48.4%)	45 (57.7%)	
Small bowel	58 (33.9%)	34 (36.6%)	24 (30.8%)	
Colon and rectum	6 (3.5%)	6 (6.4%)	0	
Ill-defined intra-abdominal site	13 (7.6%)	5 (5.4%)	8 (10.2%)	
Peritoneum	1 (0.6%)	1 (1.1%)	0	
**Death risk level**				
Very high	31 (18.1%)	19 (20.4%)	12 (15.4%)	0.707
High	77 (45.0%)	41 (44.1%)	36 (46.1%)	
Intermediate	22 (12.9%)	10 (10.8%)	12 (15.4%)	
Low	26 (15.2%)	14 (15.0%)	12 (15.4%)	
Very low	15 (8.8%)	9 (9.7%)	6 (7.7%)	

**Table 2 cancers-15-02994-t002:** Crude and adjusted incidence rates (per 100,000 inhabitants/year) of Gastro-Intestinal Stromal Tumours by sex, year of diagnosis, and Joensuu’s risk level (95% CI). Period 2001–2015, Region of Murcia, Spain.

	Crude Rate	Adjusted Rate to EU76 (95% CI)	Adjusted Rate to EU13 (95% CI)	Adjusted Rate to World Population (95% CI)
**Sex:**				
Total	0.82	0.75 (0.64–0.88)	1.05 (0.90–1.22)	0.53 (0.45–0.62)
Men	0.88	0.87 (0.69–1.07)	1.22 (0.97–1.50)	0.60 (0.48–0.75)
Women	0.76	0.65 (0.51–0.82)	0.92 (0.73–1.15)	0.46 (0.36–0.59)
**Period:**				
2001–2005	0.52	0.54 (0.37–0.77)	0.69 (0.47–0.97)	0.40 (0.27–0.57)
2006–2010	0.55	0.52 (0.37–0.72)	0.72 (0.51–0.99)	0.37 (0.26–0.53)
2011–2015	1.33	1.15 (0.92–1.41)	1.66 (1.35–2.03)	0.79 (0.63–0.98)
**Risk level:**				
Very high	0.15	0.14 (0.10–0.20)	0.20 (0.13–0.28)	0.10 (0.07–0.14)
High	0.37	0.35 (0.27–0.43)	0.47 (0.37–0.59)	0.24 (0.19–0.31)
Intermediate	0.10	0.09 (0.06–0.15)	0.13 (0.08–0.20)	0.07 (0.04–0.11)
Low/Very low	0.19	0.16 (0.11–0.22)	0.24 (0.17–0.38)	0.11 (0.08–0.16)

**Table 3 cancers-15-02994-t003:** Net survival and 95% CI of patients diagnosed of Gastro-Intestinal Stromal Tumours at 1, 3, and 5 years of follow-up, by sex. Overall survival at 5 years. Period 2001–2015, Region of Murcia, Spain.

	Net Survival (%). 95% CI			Observed Survival at 5 Years (%). 95% CI
	1 Year	3 Years	5 Years	
**Sex**				
Total	89.45 (84.45–94.46)	82.74 (76.16–89.32)	76.97 (68.16–85.55)	68.42 (60.88–74.81)
Men	85.79 (78.19–93.39)	75.68 (65.75–85.62)	71.44 (60.10–82.79)	62.37 (51.70–71.32)
Women	93.34 (87.37–99.31)	91.21 (83.10–99.32)	83.70 (73.01–94.40)	75.64 (64.51–83.71)
**Period**				
2001–2005	91.26 (81.57–100.00)	84.92 (71.12–98.72)	72.31 (55.25–89.38)	66.67 (47.94–79.96)
2006–2010	84.06 (72.27–95.84)	84.24 (71.46–97.02)	79.32 (62.72–95.92)	67.50 (50.70–79.66)
2011–2015	90.74 (84.37–97.12)	80.86 (71.71–90.02)	77.71 (67.17–88.26)	69.39 (59.23–77.49)
**Age, years**				
From 15 to 39	*	*	87.84 (66.31–100.00)	87.50 (38.70–98.14)
From 40 to 64	93.03 (86.93–99.12)	88.10 (80.11–96.09)	80.08 (70.15–90.00)	78.26 (66.56–86.28)
More than 64	85.85 (78.04–93.66)	77.35 (66.95–87.76)	73.76 (61.43–86.08)	59.57 (48.95–68.69)
**Risk level**				
**Very high**	81.72 (67.88–95.55)	63.82 (45.92–81.71)	52.66 (33.81–71.50)	48.39 (30.18–64.41)
High	92.03 (85.58–98.49)	89.12 (80.40–97.83)	81.37 (70.02–92.71)	72.73 (61.31–81.28)
Intermediate	92.64 (80.64–100.00)	*	*	90.91 (68.30–97.65)
Low/very low	87.42 (76.45–98.40)	76.58 (62.02–91.13)	71.83 (55.07–88.58)	65.74 (55.97–73.85)

* Values not available due to lack of events in follow-up periods.

**Table 4 cancers-15-02994-t004:** Hazard ratios of death for patients diagnosed with Gastro-Intestinal Stromal Tumours. Estimated by flexible parametric model (FPM) on net survival at 5 years. Period 2001–2015, Region of Murcia, Spain.

Variables		HR	95% CI	*p*-Value
Sex	**Women**	**Ref.**		
	Men	1.71	(0.92–3.20)	0.091
Age group, years	**15–39**	**Ref.**		
	40–64	3.07	(0.38–24.68)	0.292
	65 and more	5.80	(0.73–46.05)	0.096
Period	**2001–2005**	**Ref.**		
	2006–2010	0.82	(0.37–1.86)	0.643
	2011–2015	0.65	(0.33–1.28)	0.212
Risk level	Very low/low	1.38	(0.56–3.39)	0.482
	Intermediate	0.18	(0.01–5.69)	0.333
	**High**	**Ref.**		
	Very high	2.30	(1.01–5.22)	0.047

## Data Availability

The dataset is not available on public links. It can be shared in an anonymous version upon reasonable request.

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
