# Peer review of "Incidence Trends and Main Features of Gastro-Intestinal Stromal Tumours in a Mediterranean Region: A Population-Based Study"

_cancers, 2023, doi:10.3390/cancers15112994_

Round 1
Reviewer 1 Report
The authors conducted a population-based study of GIST in a region in the South-East of Spain. They elucidated clinicopathological features, incidence, and prognosis of GIST in this area. Although this is an essential investigation where GIST is one of the rare cancers, some revisions might be needed for publication.
Major comments:
l All patients have an evaluation of Joensuu’s Risk Level which is usually evaluated for resected GIST. In a real-world setting, there are patients with GIST who have no indication for surgical resection due to metastatic disease. Thus, this study may preclude those patients with metastatic disease. To assess epidemiological features, it is recommended that all patients with GIST are included in the study.
Minor comments:
l According to the previous report (Cancer Epidemiol. 2016 Feb;40:39-46), the prevalence of very low/low risk of GIST is approximately 50%. On the contrary, a prevalence of very low/low risk of GIST was around 25% in this study. Is there any reasonable association explaining the relevant differences in prevalence between the two reports? In addition, are there any temporal differences in prevalence in this study?
l The prognosis of low/very low-risk patients was poor contrary to expectations in this study. The possible reasons are documented in the Discussion section (Line 238–248). Those reasons should be presented in the Result section. In addition, documented reasons are ambiguous for explaining the poor prognosis of low/very low-risk patients. The addition of types of cancers, disease status, and treatment options will be helpful for understanding.
Author Response
Thank you so much for the review of our manuscript and your comments.
Major comments:
l All patients have an evaluation of Joensuu’s Risk Level which is usually evaluated for resected GIST. In a real-world setting, there are patients with GIST who have no indication for surgical resection due to metastatic disease. Thus, this study may preclude those patients with metastatic disease. To assess epidemiological features, it is recommended that all patients with GIST are included in the study.
Regarding the major comment, we would like to clarify that patients with metastatic GIST at diagnosis were indeed included in the study; they were assigned a Joensuu´s high risk level. All of the patients with a GIST diagnosis has immunohistochemistry and/or molecular biology techniques performed, even if it was only done on samples from the metastatic tissue instead of primary tumour.
Our decision to integrate metastatic GIST in the high risk level group is supported, for example, by Joensuu’s (et al.) 2006 paper titled: “Gastrointestinal stromal tumor (GIST)”. Quote: “The Scandinavian Sarcoma Group/AIO study randomises patients with a high risk or very high risk or recurrence (defined as tumour rupture or limited metastatic disease completely removed at surgery) to receive imatinib.”
Minor comments:
l According to the previous report (Cancer Epidemiol. 2016 Feb;40:39-46), the prevalence of very low/low risk of GIST is approximately 50%. On the contrary, a prevalence of very low/low risk of GIST was around 25% in this study. Is there any reasonable association explaining the relevant differences in prevalence between the two reports? In addition, are there any temporal differences in prevalence in this study?
The reviewer is right that prevalences in our study largely differ from those of Søreide et al. However, in the mentioned report, which took into account population-based studies from diverse continents, there was only one previous study from our own country (Rubio et al.) that reported risk stratification; its values are fairly similar to ours, so it is possible than the relative proportion of risk levels is highly dependant on the countries considered due to biological and environmental differences and different clinical guides that promote an earlier diagnosis of the smallest GIST.
Despite differences in structural biological, sociodemographic or health-related factors across countries may partially account for the differences found, it is worth noting that in our Region there was a relevant, statistically significant (p = 0.006), increase in the proportion of lower risk GIST in the last quinquennium (2010-2015): 33.7 % vs 9.1 and 7.5% in previous periods. This can be attributed to greater awareness from physicians of the necessity to address surgically the smaller lesions suggestive of GIST in imaging techniques and the improve of the techniques themselves.
l The prognosis of low/very low-risk patients was poor contrary to expectations in this study. The possible reasons are documented in the Discussion section (Line 238–248). Those reasons should be presented in the Result section. In addition, documented reasons are ambiguous for explaining the poor prognosis of low/very low-risk patients. The addition of types of cancers, disease status, and treatment options will be helpful for understanding.
The reviewer is right that those data must had been presented in the Result section, which we have already corrected. A thorough analysis of the variables suggested by the reviewer is a good point to deal with but it was beyond the scope of the present study. This issue will be take into consideration for the following studies.
Reviewer 2 Report
The authors conducted an epidemiological study on patients with gastro-intestinal stromal tumors in a small Mediterranean region.
Although the study has a clear limitation in its sample size, the follow-up seems to be sufficient and provides knowledge about other studies previously published by other authors in the field.
My main concern lies in the way of presenting the results. Given the small sample size, the tests lack sufficient contrast power to demonstrate significant differences, therefore it is not the main objective of the study, which must be addressed in a fundamentally descriptive manner. The results of the longitudinal study were reported only using tables, missing the detail of the longitudinal study provided by the survival curves.
Kaplan-Meier survival curves should be provided, detailing the number of patients at risk in the X-axis, stratified by sex, period, age, risk level and primary site of tumour.
Author Response
Answers to reviewer 2
We would like to thank the reviewer for their kind comments and very useful suggestions. The reviewer is totally right about the improvement that for our work will imply the presentation of Kaplan-Meier curves and we agree also with the specific format suggested for these. Therefore, we have added them to the draft and we also modify and expand the Methods section in order to make it clearer.
Round 2
Reviewer 1 Report
In the previous review, I pointed out that patients with metastatic GIST who are not candidate for surgery can not be evaluated by Josensuu's risk classification.
This classificaiton is made for considering risk of recurrence after surgical resection. In my understanding, this classiciation should not be adopted for metastatic disease because we can not evalutate risk of recurrence in those with metastatic disease.
The authors used an original article for arguing justification; however, target population in the article are after completely resection. Thus, patients without surgical indication are not included in this study.
Thus, the authors need to separete resectable GIST and non-resectable GIST for their analysis.
Reviewer 2 Report
All my concerns have been addressed by the authors
Author Response
The authors wish to thank you again for your appreciated comments.
Best regards.
Round 3
Reviewer 1 Report
Can metastatic cases be defined as having a worse prognosis than tumor rupture cases? Also, I believe that the Joensuu classification should be used as a recurrence risk, and it is inappropriate to use it as a risk of death.
Author Response
Answers to reviewer comments, 3rd round:
Can metastatic cases be defined as having a worse prognosis than tumour rupture cases?
Metastatic cancer disease have other defining characteristics beside their prognosis and a tumour rupture is not synonym for metastases. In the case of GIST, metastatic disease does imply a worse prognosis than tumour rupture alone; for example, in current TNM Classification for GIST, the 4th Stage, which is defined exclusively by the presence of metastatic disease, distant or on regional lymph nodes, has the worse prognosis. While establishing the TNM classification, the authors did consider tumour rupture, but despite having a prognostic value for Disease Free Survival (DFS), it was discarded for use as an Overall Survival (OS) predictor.
Ref: AJCC Cancer Staging Manual, 8th Edition.
Also, I believe that the Joensuu classification should be used as a recurrence risk, and it is inappropriate to use it as a risk of death.
Our study was not designed for evaluation of the risk of “recurrence” as defined by Joensuu, or any other classification. Instead we tried to make a description of the characteristics at diagnosis and survival rates; in doing so we stratified by several factors, among others, the risk of recurrence, since it is correlated to death risk. Other studies, including two from the references in our draft, have used recurrence risk classifications (from Joensuu and other authors) to do stratification in survival analysis (Refs. 23, 44).
On the other hand, Joensuu classification was proposed in 2008 as a modification of previous NIH classification in order to identify patients that, after surgery, will benefit the most from adjuvant treatment, focusing in Disease Free Survival; nonetheless Joensuu included both Overall Survival and Disease Free Survival outcomes in his article bibliography for the purpose of validating NIH classification.
In his article can be read: “The NIH classification was based on consensus opinion rather than on actual clinical data, but accumulating evidence now supports its true prognostic value. As shown in table 2, retrospective analyses of small GIST cohorts support the NIH classification, showing that large tumor size and high mitotic rate are generally associated with poor prognosis after surgical resection.” In the text at the bottom of the supporting graphic for this statement it is written: “Fig. 2 Estimated survival by NIH risk classification group across 6 cohorts.”
Ref: Risk stratification of patients diagnosed with gastrointestinal stromal tumor. Hum Pathol . 2008 Oct;39(10):1411-9. doi: 10.1016/j.humpath.2008.06.025
In the article from 2011: “Validation of the Joensuu risk criteria for primary resectable gastrointestinal stromal tumour – The impact of tumour rupture on patient outcomes”, being Joensuu one of the authors, can be read:
“Overall survival: The Joensuu classification risk categories were strongly associated with OS. The 5-year OS rates of patients with high-risk GIST was 78% (95% CI, 72–83%), whereas patients with intermediate, low and very low risk GIST had 97% (95% CI, 91–100%), 97% (95% CI, 91–100%) and 100% 5-year OS, respectively (P < 0.0001).”
Ref: Validation of the Joensuu risk criteria for primary resectable gastrointestinal stromal tumour – The impact of tumour rupture on patient outcomes. European Journal of Surgical Oncology (EJSO), 2011, Volume 37, Issue 10, Pages 890-896, ISSN 0748-7983,